# Evaluation of Acoustic Waves in Acousto-Optical Devices by Ultrasonic Imaging

**DOI:** 10.3390/ma15051792

**Published:** 2022-02-27

**Authors:** Sergey A. Titov, Alexander S. Machikhin, Vitold Ed. Pozhar

**Affiliations:** Scientific and Technological Center of Unique Instrumentation, Russian Academy of Sciences, 15 Butlerova, 117342 Moscow, Russia; np@ntcup.ru (A.S.M.); vitold@ntcup.ru (V.E.P.)

**Keywords:** acousto-optic crystals, propagation of acoustic waves, non-destructive testing, impulse acoustic microscopy

## Abstract

The structure of the acoustic field defines the key parameters of acousto-optical (AO) devices. To confirm their compliance with the expected values in the presence of multiple real factors, AO crystalline cells require accurate experimental investigation of the acoustic field after being totally assembled. For this purpose, we propose to detect and quantify all the acoustic waves propagating in AO cells using an impulse acoustic microscopy technique. To validate this approach, we have analyzed both theoretically and experimentally the modes, amplitudes, propagation trajectories, and other features of the ultrasonic waves generated inside an AO modulator made of fused quartz. Good correspondence between theoretical and experimental data confirms the effectiveness of the proposed technique.

## 1. Introduction

Acousto-optic (AO) interaction is a physical effect, which is used for modulation, deflection, and spectral and spatial filtration of electromagnetic radiation [1,2]. As they are compact, monolithic and free of moving components, AO devices are now widespread in industrial, biomedical and scientific applications [3,4,5]. The physical principle of most AO instruments consists of Bragg diffraction of light by ultrasound in crystalline media. The tuning of ultrasound power and frequency enables smooth and accurate control of amplitude, propagation direction, polarization and other parameters of light waves.

The structure of induced acoustic fields defines the performance of AO devices to a large extent. Attenuation, divergence, walk-off and other features of ultrasonic waves inevitably influence the key characteristics of AO diffraction: efficiency, signal-to-noise ratio, light beam distortions, etc. Precise theoretical consideration and modeling give a chance to predict and protect the characteristic degradation at the design stage, but the real structure of an acoustic field may only be revealed experimentally. It is barely possible to reveal and quantify crystal inhomogeneity and inner defects, heat generation by the piezoelectric transducer, multiple reflections of ultrasound from AO cell facets, and many other factors without experiments [6,7,8]. Therefore, it is essential to carefully examine each assembled AO cell and certify it in terms of the real acoustic field structure.

The Schlieren method has been adopted as a standard optical technique for imaging acoustic fields in homogeneous transparent media including AO crystals [1,9,10] since it provides a two-dimensional projection image formed due to the diffraction of light on an acoustic beam.

In the interferometric [11,12] and holographic [13] techniques, the spatio-temporal distribution of the probing wide laser beam is recorded by a digital camera. Since the output signal changes at the frequency of the sonic radiator, these techniques are not applicable to high-frequency AO devices. Laser beam scanning also enables visualization of acoustic fields [14,15,16,17]. In [14,15,16], the laser beam diameter is smaller than the sound wavelength; therefore, this scheme is applicable only to low-frequency fields. In [17], the shape of the diffracted laser beam allows evaluation of high frequency ultrasonic wave in solid media. 

In all these methods [9,10,11,12,13,14,15,16,17], the measurement results are in fact averaged over the entire optical path through the sound field. Therefore, in practice, they are effective for the analysisof simple sound fields that have plane or symmetrical wave fronts. In real AO devices, many waves of various types with different directions and amplitudes can propagate. To obtain three-dimensional distributions of ultrasonic fields, tomographic methods based on the interferometric [18,19,20] or advanced Schlieren schemes [21,22,23] are applicable. In this case, data acquisition from different angles is necessary, which is hardly possible for AO devices due to their design features.

The acoustic field structure in crystal influences the wave displacement distribution on the facet of an AO cell located opposite or next to the transducer. Such measurement is available for the laser ultrasonic technique [24] but needs a highly reflective facet, which is normally absent. The gold standard for measuring the acoustic field is the use of needle hydrophones [25,26]. However, they cannot be placed directly on the AO cell facet and have low sensitivity due to their small dimensions. In this paper, we propose to use a focused immersion ultrasonic transducer with the focus placed on the cell surface. This scheme is typical for the receiver of a pulse acoustic scanning microscope [27], which has a high signal-to-noise ratio and spatio-temporal resolution.

In this study, we intend to validate this approach by studying an AO modulator made of fused quartz in order to determine both theoretically and experimentally the modes, amplitudes, propagation trajectories and other features of the ultrasonic waves inside. Below, we describe the experimental setup, the theoretical model of acoustic field structure and the experimental data, then discuss the results and demonstrate good correspondence between theoretical prediction and acoustic microscopy data that confirm the effectiveness of the proposed technique.

## 2. Experimental Setup and Technique

To confirm the feasibility of this approach, we applied it to testing an AO cell 1 made of optical fused quartz (Figure 1). It has the shape of a straight prism with a thickness of 12 mm. To generate the longitudinal acoustic waves in the cell, a lithium niobate ultrasonic transducer 2 is attached to the bottom plane. To avoid the formation of standing acoustic waves, the upper plane of the cell is inclined at the angle α = 6.5°. The transducer consists of two 17.5 × 3.5 mm^2^ sections separated by a 4 mm gap. The central frequency is 50 MHz while the frequency range is 30 MHz. 

The upper part of the cell is inserted into a cuvette 3 filled with an immersion liquid (water). The ultrasonic waves generated by the piezoelectrical transducer 2 propagate inside the AO cell 1 and partially penetrate into the immersion liquid. These waves are detected by a focused piezotransducer 4 with a central frequency of 50 MHz and an angular aperture of 15°. The focus of the transducer 4 located at the quartz–water interface is mechanically translated along this plane to record the spatial distribution of the waves. 

The ultrasonic data acquisition system of the experimental setup consists of the blocks typical for a scanning acoustic microscope [27,28]. To separate responses of different waves, the pulsed mode is used. The pulser–receiver 5 (5073PR, Panametrics–NDT Inc., Waltham, MA, USA) generates short electrical pulses to feed the AO piezotransducer 2 and amplifies the weak output signals of the transducer 4. The signals are then processed by the analog-to-digital converter 6 (FM412x500M, Insys Inc., Moscow, Russia) at a sampling rate of 500 MHz and a resolution of 12 bits. The mechanical movement of the focused transducer is implemented by the motorized translation stage 7 (8MT167-100, Standa Ltd., Vilnius, Lithuania). At each position of the transducer, the acquired signal is averaged over 16 series in order to increase the signal-to-noise ratio and then is recorded as a function of the wave propagation time *t* within temporal widows of 4 µs. We acquired the signals at different positions separated by a spatial period of 0.1 mm. The travel distance is set to 44 mm to cover the entire top facet of the AO cell. Recording time of one scan is about 4 s. To obtain the delayed multiple reflection responses, the scan is repeated several times with different time window settings and signal gain.

There is a variety of ultrasonic waves propagating inside the AO cell (Figure 1), both longitudinal and transverse, while in the immersion liquid there are only longitudinal ones. Normally, the transducer generates a plane longitudinal wave *L*, but a weak transverse wave *T* may also appear. Both waves undergo reflection and mode conversion at the upper solid–liquid interface and partially penetrate the water. Wave *L* produces a longitudinal wave LL and a transverse wave LT, which transform at the bottom plane into two longitudinal waves, LLL and LTL, and two transverse waves, LLT and LTT. 

The applied technique presumes detection of the water-penetrating waves by a scanning transducer focused on the upper face of the AO cell. These signals, together with information about the acoustic properties of the cell and immersion liquid are necessary to calculate the amplitudes of upward and downward waves in the AO cell. Then, we may evaluate the amplitudes of the waves inside the AO device using the characteristics of the acoustic absorber. To verify the theoretical data, we detect the waves on the clear part of the upper face partly covered with the absorber.

## 3. Theoretical Model

In the theoretical analysis described below, we have made the following assumptions. Since the dimensions of the transmitting transducer are much larger than the ultrasound wavelength, the divergence is rather small, the wave fronts in quartz and water are approximately plane and the ray approximation is applicable. As the acoustic attenuation in quartz is small, the amplitudes in the bottom face and near the quartz-water interface are equal. Therefore, the well-known formulas for the reflection, transmission and mode conversion coefficients for plane waves at the solid–liquid and solid–solid interfaces are valid [29]. All the coefficients are real because the angles of incidence do not exceed the critical values. The angles between the wave vectors in water and the axis of the receiving transducer are significantly less than its angular aperture. The distance between the transducer and the interface is constant and, therefore, the sensitivity of the receiver does not depend on the wave propagation direction.

In theoretical analysis, we use the symbols η, µ, ξ, κ = *L*, and *T* to denote waves. Let *p*_η_ be the amplitudes of the waves *L* and *T* radiated by the transducer (Figure 1). The four waves reflected back at the upper interface have amplitudes *p*_ηµ_. In the next step, eight waves with the amplitudes *p*_ηµξ_ propagate in the upward direction.

The incidence angles of the primary waves *p*_η_ on the upper surface are equal to the inclination angle α (Figure 2). Incidence angles γ_η__µ_ and θ_η__µ__ξ_ of reflected waves are:(1)$$\gamma_{\eta \mu }=\beta_{\eta \mu }+\alpha \cdot \theta_{\eta \mu \xi }=\delta_{\eta \mu \xi }+\alpha .$$

The angles of reflection satisfy Snell’s law:(2)$$\sin\left(\beta_{\eta \mu }\right)=\frac{C_{\mu }}{C_{\eta }}\sin\left(\alpha \right),\;\sin\left(\delta_{\eta \mu \xi }\right)=\frac{C_{\xi }}{C\mu }\sin\left(\gamma_{\eta \mu }\right),$$
where *C_L_* = 5960 m/s and *C_T_* = 3760 m/s are velocities of the longitudinal and transverse waves in fused quartz, respectively [27]. Refracted waves in water propagate at angles φ_η_ and φ_ηµξ_:(3)$$\sin\left(\varphi_{\eta }\right)=\frac{C_{W}}{C_{\eta }}\sin\left(\alpha \right),\;\sin\left(\varphi_{\eta \mu \xi }\right)=\frac{C_{W}}{C_{\xi }}\sin\left(\theta_{\eta \mu \xi }\right),$$
where *C_W_* = 1485 m/s is sound velocity in water.

The spatio-temporal signals received by the focused transducer may be described as
(4)$$S_{\eta }\left(x,t\right)=a_{\eta }w\left(t-t_{\eta }\left(x\right)\right),\;S_{\eta \mu \xi }\left(x,t\right)=a_{\eta \mu \xi }w\left(t-t_{\eta \mu \xi }\left(x\right)\right),$$
where *x* and *t* are the scanning coordinate and time, *w*(*t*) is the impulse response of the experimental setup, *a*_η_ and *a*_ηµξ_ are the amplitudes of the compression waves in water generated by corresponding modes *p*_η_ and *p*_ηµξ_, and *t*_η_ and *t*_ηµξ_ are their delays. The amplitudes *a*_η_ and *a*_ηµξ_ are proportional to the amplitudes of the waves penetrated into the immersion liquid:(5)$$a_{\eta }=p_{\eta }T_{\eta }\left(\alpha \right)h,$$
where *T*_η_(α) are the transmission coefficients at the solid–liquid interface and *h* is the detector sensitivity coefficient. For *p_L_* = 1, this coefficient is equal to
(6)$$h=\frac{a_{L}}{T_{L}\left(\alpha \right)}.$$

Then, the relative amplitude of the transverse wave *T* can be found using measured values of *a_L_* and *a_T_* as
(7)$$p_{T}=\frac{a_{T}T_{L}\left(\alpha \right)}{a_{L}T_{T}\left(\alpha \right)}.$$

The amplitudes of the reflected waves may be found as follows:(8)$$p_{\eta \mu }=p_{\eta }R_{\eta \mu }\left(\alpha \right),$$
where the reflection coefficient *R*_ηµ_(α) is determined by the acoustic properties of fused quartz and water. Substitution of (5) and (6) in Equation (7) gives
(9)$$p_{\eta \mu }=\frac{a_{\eta }R_{\eta \mu }\left(\alpha \right)}{hT_{\eta }\left(\alpha \right)}=\frac{a_{\eta }R_{\eta \mu }\left(\alpha \right)T_{L}\left(\alpha \right)}{a_{L}T_{\eta }\left(\alpha \right)}.$$

Since
(10)$$a_{\eta \mu \xi }=p_{\eta \mu \xi }T_{\xi }\left(\theta_{\eta \mu \xi }\right)h,$$
the amplitudes *p*_ηµξ_ can be estimated in a similar way using the measured values *a*_ηµξ_: (11)$$p_{\eta \mu \xi }=\frac{a_{\eta \mu \xi }}{T_{\xi }\left(\theta_{\eta \mu \xi }\right)h}=\frac{a_{\eta \mu \xi }T_{L}\left(\alpha \right)}{a_{L}T_{\xi }\left(\theta_{\eta \mu \xi }\right)}.$$

After the absorber is attached, the direct measurement of the wave amplitudes is not feasible. Nevertheless, the amplitudes of the primary waves *p*_η_ remain the same, whereas the amplitudes of the reflected waves in the cell covered by absorber *p*^*^_ηµ_, *p*^*^_ηµξ_ decrease in proportion to the reflection or mode conversion coefficient at the quartz–absorber interface *R*^*^_ηµ_(α):(12)$$p_{\eta \mu }^{*}=p_{\eta \mu }\frac{R_{\eta \mu }^{*}\left(\alpha \right)}{R_{\eta \mu }\left(\alpha \right)},\;$$
(13)$$p_{\eta \mu \xi }^{*}=p_{\eta \mu \xi }\frac{R_{\eta \mu }^{*}\left(\alpha \right)}{R_{\eta \mu }\left(\alpha \right)}.$$

The delays *t*_η_ and *t*_ηµξ_ of the signals (4) may be calculated as well as the propagation distances of the corresponding waves inside the AO cell (Figure 2). The distance *d* between the transducer and the receiving focus point F depends linearly
(14)d=L−xtanα,
where *x* is the focus position and *L* is the length of the left facet. The delays of *L* and *T* waves are proportional to the distance *x*:(15)$$t_{\eta }=t_{0\eta }-\varepsilon_{\eta }x,$$
where
(16)$$t_{0\eta }=\frac{L}{C_{\eta }},\;\varepsilon_{\eta }=\frac{\tan\alpha }{C_{\eta }}.$$

The delays *t*_ηµξ_ depend on the travel distances *d*_1_, *d*_2_ and *d*_3_ (Figure 2) in the following way:(17)$$t_{\eta \mu \xi }=\frac{d_{1}}{C_{\xi }}+\frac{d_{2}}{C_{\mu }}+\frac{d_{3}}{C_{\eta }}.$$

Since
(18)$$d_{1}=\frac{d}{\cos\delta_{\eta \mu \xi }},\;d_{3}=d_{2}\cos\gamma_{\eta \mu },\;\mathrm{and}\;d_{2}=d_{1}\frac{\cos\theta_{\eta \mu \xi }}{\cos\beta_{\eta \mu }},$$
(19)$$t_{\eta \mu \xi }=t_{0\eta \mu \xi }-\varepsilon_{\eta \mu \xi }x,$$
where
(20)$$t_{0\eta \mu \xi }=KL,\;\varepsilon_{\eta \mu \xi }=K\cdot \tan\alpha ,$$
we derive
(21)$$K=\frac{1}{\cos\delta_{\eta \mu \xi }}\left(\frac{1}{C_{\xi }}+\frac{\cos\theta_{\eta \mu \xi }}{C_{\mu }\cos\beta_{\eta \mu }}+\frac{\cos\theta_{\eta \mu \xi }}{C_{\eta }\cos\beta_{\eta \mu }}\cos\gamma_{\eta \mu }\right)$$ .

Thus, the values of delay *t*_η_ and *t*_ηµξ_ are expressed as a linear dependence on the scanning length *x* with parameters presented in Table 1. 

## 4. Experiments

In this section, we present the experimental results of acoustic pulse detection in two different conditions: (1) an AO cell with absorber fully removed and (2) a similar AO cell partly covered with absorber. We analyze the wave patterns and determine their amplitude and delays in order to estimate AO cell characteristics.

The measured ultrasonic signals are shown in Figure 3 and Figure 4 as grayscale images. In these images, the signal value is encoded by the gray levels and is presented as a function of the retarded time τ:(22)$$\tau =t_{0}-\varepsilon x$$

The value of slowness ε is matched to compensate for the spatial dependence on τ. The *t*_0_ values were then measured using the positions of the maxima of the ultrasonic pulse envelopes. As the variable part of the propagation time is compensated, the responses *S*(*x*,τ) look like horizontally oriented patterns. The measured values *t*_0_ and ε are also presented in Table 1. There is a good agreement between the experimental and calculated values that confirms the correctness of the theoretical model. 

The response *S_L_*(*x*,τ) (Figure 3a) is generated by the main longitudinal wave *L*. There are two horizontal components, P_1_ and P_2_, and a set of wavelets E adjacent to P. The size and position of the components P_1_ and P_2_ correspond to the size and position of the sections of the transmitting transducer indicated by Tr_1_ and Tr_2_. The wavelets E are radiated by the transducer’s edges. This wave pattern is typical for a flat piston transducer [30]. 

The values of arrival time *t*_0*T*_ and coefficient ε*_T_* of the transverse wave *T* are greater than those for the longitudinal wave *L* (*t*_0*L*_ and ε*_L_*), since *C_T_* < *C_L_*. The response *S_T_*(*x*,τ) from wave *T* also contains two components, P_1_ and P_2_, the sizes and positions of which coincide with the transmitting transducer aperture (Figure 3b). Wave *T* is much weaker than wave *L* and, therefore, random noise and artifacts (marked by A in Figure 3b) generated by some unwanted echoes inside the measurement setup are present in the image. 

The LLL wave formed by triple passes of the longitudinal wave is shown in Figure 4a. There is a lateral displacement Δ*x* of the response due to the slope of the upper facet. The value of Δ*x* may be estimated from the ray model (Figure 2):(23)$$\Delta x\approx L\left(\tan\delta_{\eta \mu \xi }+\tan\gamma_{\eta \mu }\right)$$

The experimentally evaluated displacement is consistent with the theoretical estimate Δ*x* ≈ 12 mm. The component P_1_ is short because the left part of the ultrasonic beam from the section Tr_1_ is bounded by the left facet of the AO cell. In the wavelet P_2_, there is irregularity G formed by the reflection of the wave LL from the gap between two sections of the transducer on the bottom face instead of the reflection from the transducer’s surface.

The measured spatio-temporal signal *S_LTT_*(*x*,τ) is produced by the mode conversion of the longitudinal wave *L* at the upper interface (Figure 4b). The structure of this response is similar to *S_LLL_*(*x*,τ), but the displacement Δ*x* ≈ 9.5 mm is less due to the fact that the angles δ_LTT_ and γ_LTT_ are less than δ_LLL_ and γ_LLL_, respectively (Table 1).

To determine the amplitudes of the recorded signals, we estimated the maximal values of the envelopes of the ultrasonic pulses. Due to the irregularity of the transducer radiation efficiency, diffraction effects and artifacts, some spatial variations are present in the received responses. To reduce the measurement uncertainty, we calculated the mean and standard deviation of the envelope maximal values within a certain spatial window. The window is set in the range of 30 < *x* < 40 mm for *L* and *T* waves (Figure 3). For the rest of the waves (Figure 4), a window of 18 < *x* < 28 mm was used to compensate for the spatial displacement Δ*x*. The average amplitudes *a*_η_ and *a*_ηµξ_ are normalized by the amplitude *a_L_*. Their relative standard deviations σ_η_ and σ_ηµξ_ are presented in Table 2. To assess the measurement error of the experimental setup, the random noise level is estimated as the root mean square of the recorded signal in the areas without wave responses and artifacts. This noise value is 0.3%, which is much less than the standard deviations of the amplitudes.

To evaluate the influence of the absorber, we used another AO cell of the same design. The cell has the absorber installed on the upper facet. The absorber is made from epoxy resin and covers a part of the facet as shown in Figure 5. The left edge of the absorber is located at *x*_0_ ≈ 30 mm. Therefore, the signal *S_LLL_*(*x*,τ) is damped at the interval [*x*_1_, *x*_2_] ≈ [18, 28] mm due to the decrease of the reflection coefficient *R_L_* in the presence of the absorber. The ratio of amplitudes measured for the quartz–absorber and quartz–water configurations is 0.75.

## 5. Discussion

The wave amplitudes in quartz *p_T_*, *p*_ηµ_ and *p_L_*_µξ_ are determined from the ones measured in water *a*_η_ and *a*_ηµξ_ (Table 2) using Equations (7), (9) and (11). Well-known formulas [29] allows calculation of the transmission coefficients *T*_η_(α) and *T*_ξ_(θ_ηµξ_), reflection and mode conversion coefficients *R*_ηµ_(α) for the fused quartz–water interface. These coefficients are shown in Figure 6 as functions of the refraction angle φ in water. Their values at the angles φ_η_, φ_ηµξ_ are presented in Table 2 as well as the obtained relative amplitudes *p*_η_, *p*_ηµ_ and *p*_ηµξ_.

We have discovered that the calculated relative amplitude of the direct transverse wave *p_T_* = 0.078 is significant. Thus, the ultrasonic transducer generates a non-negligible transverse wave along with the regular longitudinal wave. This unwanted wave propagates in the same direction and cannot be attenuated by the absorber. 

Among the reflected waves, the LL wave with an amplitude of *p_LL_* = 0.77 is the largest. The waves TL and TT, generated at the upper interface by the mode conversion of the wave *L* and reflection of the transverse wave *T*, are much weaker. Therefore, only the propagation of LL and LT waves is considered below.

We should note that the amplitudes *p*_ηµξ_ may be calculated using the determined values *p*_ηµ_ and the reflection coefficients at the bottom interface. However, the design of the transmitting transducer is rather complicated and it is difficult to find the reflection coefficients from its surface. In addition, the reflectivity depends on the electrical load of the transducer. Thus, this approach is not reliable and barely practically realizable.

In the presence of the absorber, the amplitudes *p**_η_ of the direct waves do not change, while the amplitudes *p**_ηµ_ and *p**_ηµξ_ become smaller. These amplitudes are estimated from the amplitudes *p*_ηµ_ and *p*_ηµξ_ via Equations (12) and (13) (Table 2). We calculated the required reflection and mode conversion coefficients *R^*^*_ηµ_(α) under the assumption that the absorber is a solid medium with a density of 1150 kg/m^3^ and velocities of longitudinal and transvers waves of 2650 m/s and 1100 m/s, respectively, values which are typical for epoxy resin [31]. The coefficients *R^*^*_ηµ_ presented in Figure 6 and Table 2 are obtained based on well-established technique [29]. In the experimental data, the ratio *p**_LLL_/*p*_LLL_ is 0.75, which is quite close to the value 0.79 calculated from Table 2. We tested the proposed method on a model absorber and confirmed its applicability. Though the estimated amplitudes *p**_ηµ_ and *p**_ηµξ_ are small, they are not negligible and their effect may be significant for the detailed analysis of the AO device characteristics.

## 6. Conclusions

We have shown that impulse acoustic microscopy is quite informative tool for quantitative characterization of the acoustic field in AO cells. It allows to define the modes, amplitudes, propagation trajectories and other features of the ultrasonic waves propagating in the crystal even after multiple reflections. This information is highly important in practice as it enables the evaluation of the correctness of AO cell design including the cut and facet angles, efficiency of the ultrasound piezotransducer and absorber functioning, etc. Being non-destructive and highly sensitive, impulse acoustic microscopy might be effective for quite fast experimental validation of theoretical estimations and numerical modeling results as well as for accurate certification of AO devices.

## Figures and Tables

**Figure 1 materials-15-01792-f001:**
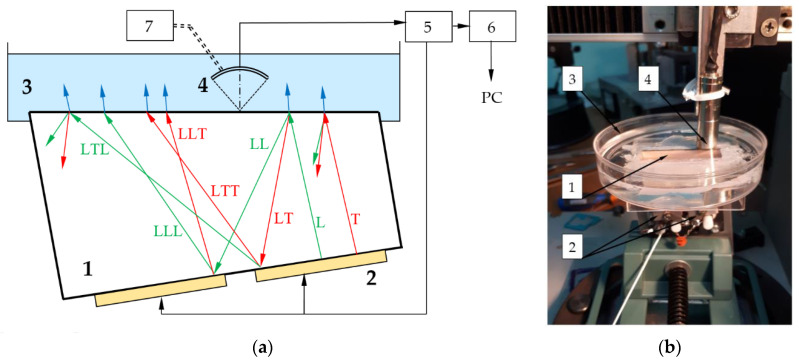
Scheme (**a**) and appearance (**b**) of the experimental setup: 1—AO cell; 2—emitting piezotransducer; 3—cuvette with immersion liquid; 4—receiving piezotransducer; 5—pulser-receiver; 6—analog-to-digital converter; 7—motorized translator. Green and red arrows show longitudinal (*L*) and transverse (*T*) waves propagating in AO cell, blue arrows correspond to longitudinal waves in liquid.

**Figure 2 materials-15-01792-f002:**
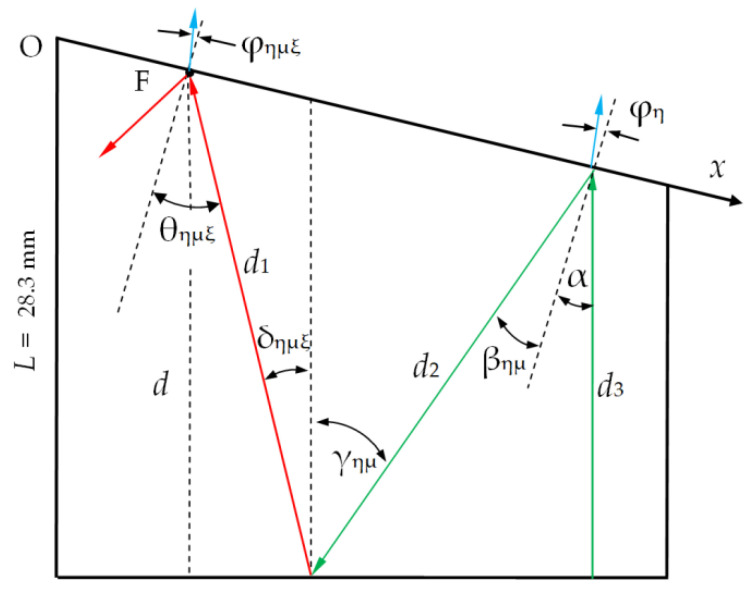
Ray model of the acoustic wave propagation in the AO cell.

**Figure 3 materials-15-01792-f003:**
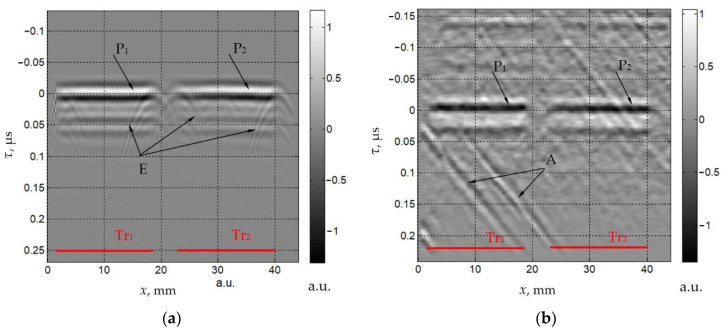
Measured spatio-temporal signals *S_L_*(*x*,τ) (**a**) and *S_T_*(*x*,τ) (**b**).

**Figure 4 materials-15-01792-f004:**
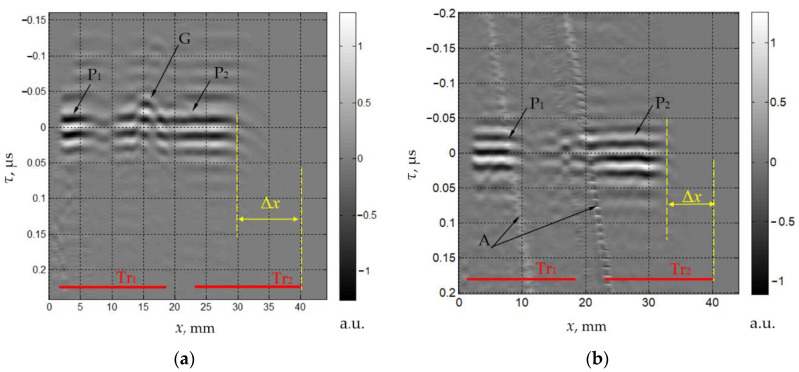
Measured spatio-temporal signals *S_LLL_*(*x*,τ) (**a**) and *S_LTT_*(*x*,τ) (**b**).

**Figure 5 materials-15-01792-f005:**
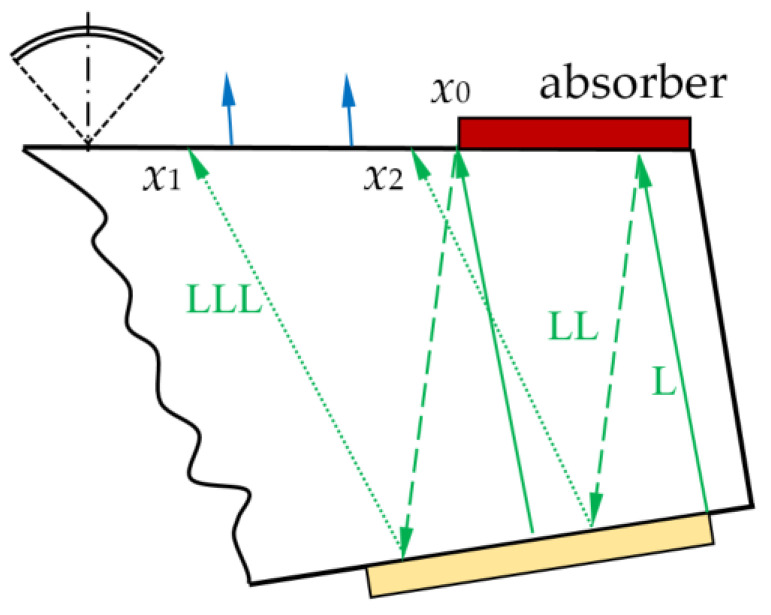
Ray propagation in AO cell with absorber.

**Figure 6 materials-15-01792-f006:**
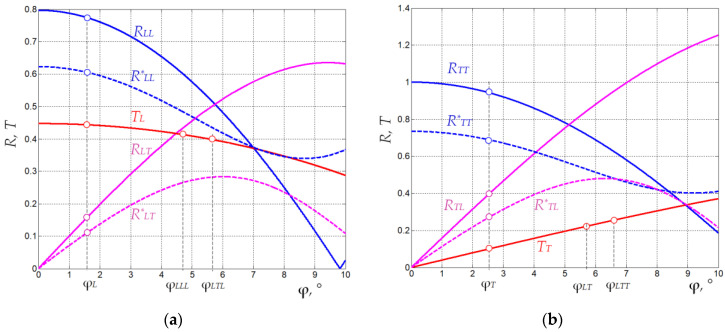
(**a**) Reflection coefficients *R_LL_*, *R^*^_LL_*, *R_TT_* and *R^*^_TT_*, transmission coefficients *T_L_*, and *T_T_*, mode conversion coefficients *R_LT_*, *R^*^_LT_*, *R_TL_* and *R^*^_TL_* for the interfaces fused quartz–water (solid lines) and fused quartz–absorber (dashed lines) vs refraction angle in water φ: longitudinal (**a**) and transverse (**b**) waves in quartz.

**Table 1 materials-15-01792-t001:** Calculated and experimental values of angular, temporal and slowness parameters of wave propagation.

Wave	Calculation						Experiment	
	φ_η_, φ_ηµξ_, °	γ_ηµ_, °	δ_ηµξ_, °	θ_ηµξ_, °	*t*_0η_, *t*_0ηµξ_, µs	ε_η_, ε_ηµξ_, ns/mm	*t*_0η_, *t*_0ηµξ_, µs	ε_η_, ε_ηµξ_, ns/mm
η = *L*	1.62	-	-	-	4.755	19.1	4.75	19.3
η = *T*	2.56	-	-	-	7.54	30	7.9	30.8
ημξ = LLL	4.77	13	13	19.5	14.02	56.4	13.95	57.5
ημξ = LLT	5.74	13	8.16	14.66	16.61	66.8	16.5	67.0
ημξ = LTL	5.70	10.6	16.9	23.5	17.2	69.2	16.8	69.1
ημξ = LTT	6.67	10.6	10.6	17.1	19.57	78.7	19.6	80.5

**Table 2 materials-15-01792-t002:** Experimental data for absorber-free AO cell and AO cell covered with ultrasound absorber: refracted angles (ϕ), normalized amplitudes (*a*, *p*), transmission (*T*) and reflection (*R*) coefficients for various modes and given inclination angle (α = 6.5°).

Wave Mode	Quartz-Water Interface (Absorber-Free AO Cell)						Quartz-Epoxy Interface (Absorber-Covered Cell)	
	Measured		φ_η_, φ_ηµξ_,°	*T*_η_ (α) *T*_ξ_(θ_ηµξ_)	*R*_ηµ_ (α)	*p*_η_, *p*_ηµ_, *p*_ηµξ_	*R^*^*_ηµ_ (α)	*p**_η_, *p**_ηµ_, *p**_ηµξ_
	*a*_η_, *a*_ηµξ_	σ_η_*,* σ_ηµξ_,%						
η = *L*	1	8.2	1.62	0.444	-	1	-	1
η = *T*	0.018	13.5	2.56	0.101	-	0.078	-	0.078
ημ = LL	-	-	-	-	0.773	0.77	0.605	0.605
ημ = LT	-	-	-	-	0.161	0.16	0.11	0.11
ημ = TL	-	-	-	-	0.401	0.031	0.27	0.021
ημ = TT	-	-	-	-	0.94	0.073	0.69	0.054
ημξ = LLL	0.39	12.5	4.77	0.413	-	0.42	-	0.33
ημξ = LLT	0.051	25	5.74	0.222	-	0.10	-	0.068
ημξ = LTL	0.074	22	5.70	0.398	-	0.083	-	0.056
ημξ = LTT	0.075	22.6	6.67	0.257	-	0.13	-	0.095

## Data Availability

The data presented in this study are available on request from the corresponding author.

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
