# Peer review of "Evaluation of Acoustic Waves in Acousto-Optical Devices by Ultrasonic Imaging"

_materials, 2022, doi:10.3390/ma15051792_

Round 1
Reviewer 1 Report
In this manuscript, the authors proposed and demonstrated a technique to detect and quantify the acoustic waves propagation by impulse acoustic microscopy. The work is generally described correctly, but there are a few things that could be described more precisely.
1. Theoretical model for acoustic wave is much mature and may not need so much attention.
2. Since the ultrasonic imaging by impulse acoustic microscopy is the key point of this work, the detailed imaging process and related experimental results should be presented in more detail.
3. For the comparison table, it is suggested to add some vertical bars to distinguish different columns.
4. The clarity of the paper may benefit from being carefully proof read and some style and gramma corrections should be made.
Author Response
REVIEWER 1
Thank you for your valuable comments. To address your remarks, we have revised the text. All changes are marked with yellow color.
(1) Theoretical model for acoustic wave is much mature and may not need so much attention.
Actually, the paper now contains very basic equations necessary for the theoretical evaluation of the acoustic waves and further comparison with the experimental values. If you find them excessive or/and unnecessary, we may transfer these calculations to Appendix or Supplemental material.
(2) Since the ultrasonic imaging by impulse acoustic microscopy is the key point of this work, the detailed imaging process and related experimental results should be presented in more detail.
We have added more details on the imaging procedure and the obtained data.
(3) For the comparison table, it is suggested to add some vertical bars to distinguish different columns.
We have added the vertical bars in both tables.
(4) The clarity of the paper may benefit from being carefully proof read and some style and gramma corrections should be made.
We have double checked the manuscript for grammatical errors and corrected them.
Reviewer 2 Report
The authors propose and validate pulse acoustic microscopy for visualizing acoustic waves in an acousto-optic cell with varied configurations of absorbers, transducers, etc. I believe this work may be moderately interesting, because the proposed method may provide additional or complementary advantages to existing techniques. However, the appeal of this method may be limited to a small subset of the community; the authors must provide improved comparison of their method with existing methods, like those listed in the introduction. Appropriate benchmarking is crucial to the impact and novelty of this work; however, as it stands, no such comparisons are made with other literature. Ultimately, I suggest major revisions are required until such changes emphasizing the novelty of the method proposed herein and comparison with existing techniques are included. Otherwise, please addresses the following minor changes:
Specific comments:
- Please fix double punctuation in line 54.
- Please correct typographical error in line 80, spacing between “waves.Normally”.
- Please correct grammar in line 86, “The applied technique presumes detection of the …”
- The authors should elaborate on how the measurement uncertainty is determined in the results shown in Table 2. Are these determined from averaging several such measurements?
- It is difficult to differentiate between measurement and calculation in Table 2 and following discussion; the authors should provide a clear and concise comparison of their experimental results to calculations.
Author Response
REVIEWER 2
Thank you for your valuable comments. To address your remarks, we have revised the text. All changes are marked with yellow color.
(1) I believe this work may be moderately interesting, because the proposed method may provide additional or complementary advantages to existing techniques. However, the appeal of this method may be limited to a small subset of the community; the authors must provide improved comparison of their method with existing methods, like those listed in the introduction. Appropriate benchmarking is crucial to the impact and novelty of this work; however, as it stands, no such comparisons are made with other literature. Ultimately, I suggest major revisions are required until such changes emphasizing the novelty of the method proposed herein and comparison with existing techniques are included.
We have revised Introduction by adding more references and emphasizing the novelty of the proposed technique in comparison with the existing ones.
(2) Please fix double punctuation in line 54.
Corrected.
(3) Please correct typographical error in line 80, spacing between “waves.Normally”.
Corrected.
(4) Please correct grammar in line 86, “The applied technique presumes detection of the …”
Corrected.
(5) The authors should elaborate on how the measurement uncertainty is determined in the results shown in Table 2. Are these determined from averaging several such measurements?
We have modified Table 2 and added additional explanation to the text
(6) It is difficult to differentiate between measurement and calculation in Table 2 and following discussion; the authors should provide a clear and concise comparison of their experimental results to calculations.
We have modified Table 2 so that now the experimental results and calculations might be distinguished clearly.
Reviewer 3 Report
Reviewer report:
The manuscript “Evaluation of acoustic waves in acousto-optical devices by ultrasonic imaging” presents an interesting work in the field of acoustic scanning. The main question addressed by the research is the study of acoustic waves in acousto-optical devices. Each such device has unique acoustic modes and these modes should be imaged to evaluate the characteristics of the particular device.
In the manuscript, detailed calculations are made and results are compared with results of impulse acoustic microscopy measurements. The comparison shows good agreement between the experiment and theory. The Paper is well written, the material is systemized and presented in logical sequence which is easy to read and understand. Conclusions are consistent with the results.
I recommend publishing the manuscript after the following revision:
It is recommended to include more works in the References section. There are a lot of related works published recently.
Author Response
REVIEWER 3
Thank you for your valuable comments. To address your remarks, we have revised the text. All changes are marked with yellow color.
(1) It is recommended to include more works in the References section. There are a lot of related works published recently.
We have revised Introduction by adding more references and emphasizing the novelty of the proposed technique in comparison with the existing ones.
Round 2
Reviewer 1 Report
Detailed modification should be presented in the response letter. Otherwise, it is not so convenient to judge what change has been made.
Reviewer 2 Report
The authors adequately addressed my previous concerns about comparing to existing methods and literature. The manuscript can be published with minor grammatical and formatting changes.
Author Response
Thank you for your comments; we have revised the text again. New changes are marked with yellow color
Round 3
Reviewer 1 Report
The authors have responded to all my concerns.